# Multiple Myeloma: Challenges Encountered and Future Options for Better Treatment

**DOI:** 10.3390/ijms23031649

**Published:** 2022-01-31

**Authors:** Srijit Das, Norsham Juliana, Noor Anisah Abu Yazit, Sahar Azmani, Izuddin Fahmy Abu

**Affiliations:** 1Department of Human & Clinical Anatomy, College of Medicine & Health Sciences, Sultan Qaboos University, Al-Khoud, Muscat 123, Oman; drsrijit@gmail.com; 2Faculty of Medicine and Health Sciences, Universiti Sains Islam Malaysia, Persiaran Ilmu, Putra Nilai, Nilai 71800, Negeri Sembilan, Malaysia; anisahyazit@gmail.com (N.A.A.Y.); drazmanisahar@usim.edu.my (S.A.); 3Institute of Medical Science Technology, Universiti Kuala Lumpur, Kuala Lumpur 50250, Selangor, Malaysia; izuddin@unikl.edu.my

**Keywords:** multiple myeloma, management, drug resistance, epigenetic, modifications

## Abstract

Multiple myeloma (MM) is a malignant hematological disease. The disease is characterized by the clonal proliferation of malignant plasma cells in the bone marrow. MM accounts for 1.3% of all malignancies and has been increasing in incidence all over the world. Various genetic abnormalities, mutations, and translocation, including epigenetic modifications, are known to contribute to the disease’s pathophysiology. The prognosis is good if detected early, or else the outcome is very bad if distant metastasis has already occurred. Conventional treatment with drugs poses a challenge when there is drug resistance. In the present review, we discuss multiple myeloma and its treatment, drug resistance, the molecular basis of epigenetic regulation, the role of natural products in epigenetic regulators, diet, physical activity, addiction, and environmental pollutants, which may be beneficial for clinicians and researchers.

## 1. Introduction

### 1.1. Multiple Myeloma

Multiple myeloma (MM) is a disease in which there is a clonal expansion of plasma cells in the bone marrow and that results in the production of a monoclonal protein and end-organ damage [1]. The first case of MM dates back to 1844, when it was reported by Samuel Solly, and the case of Alexander McBean, a well-known tradesman from London in 1850, is also worth mentioning [2]. Later, in the middle of the 19th century, Henry Bence Jones described the large amount of protein excreted by Mr. McBean [2].

MM starts as a benign condition known as monoclonal gammopathy of undetermined significance (MGUS). In high-income countries, MM is the second most prevalent hematological cancer [3]. With an annual incidence of 4.5–6 cases per 100,000 individuals, MM accounts for 1.3% of all malignancies and 15% of all hematological neoplasms [1]. In the last few decades, the incidence of MM has been alarmingly increasing, and approximately 86,000 cases are reported to occur annually all over the world [4]. MM is usually seen in older individuals and is observed to be more common in males compared to females [5]. Published studies have reported a median age at diagnosis of 72 years with a mortality rate of 4.1 per 100,000 individuals per year [6].

Various genetic abnormalities, mutations, and translocations, including epigenetic modifications such as DNA and histone methylation, and abnormal miRNA, are known to contribute to the disease’s pathophysiology [7,8,9,10]. There are different risk factors for MM, and they include obesity, chronic inflammation, pesticide or organic solvent exposure, and radiation [3]. Researchers showed a positive association between body mass index (BMI) and the incidence of MM in whites [11]. Interestingly, studies reported that high BMI and change in glucose metabolism are related to an increase in the incidence of MM [12].

Deletion 17p13, t(4;14)(p16;q32), t(14;16)(q32;q23), Metaphase deletion 13, and metaphase hypodiploid abnormalities were associated with high risk cases [13]. In a recent study published in 2020, researchers highlighted genetic and societal factors that contribute to racial and ethnic differences in the prevalence and prognosis of multiple myeloma and its precursor state [14].

One of the earliest events related to the pathogenesis of MM include IgH translocations involving the chromosome 14q32 [15]. Almost all MMs begin with an asymptomatic premalignant stage, which is known as monoclonal gammopathy of undetermined significance, i.e., MGUS [13]. In the development of MM, the bone marrow is thought to play a crucial role. In the later stages of the disease, malignant plasma cells can survive outside the bone marrow and migrate to other tissues or circulate in the peripheral blood. The disease is complicated by organ dysfunction, i.e., hypercalcaemia, renal insufficiency, anemia, and bone destruction, which is also known as the CRAB criteria [3].

### 1.2. Diagnostic Criteria for Multiple Myeloma

In MM, monoclonal plasma cells infiltrate the bone marrow, and there is secretion of monoclonal immunoglobulin, which is detected in blood or urine. To diagnose MM, the CRAB criteria are used. Organ damage is usually identified by the acronym CRAB, and this includes C—calcium elevation (>11.5 mg/dL), R—renal dysfunction (creatinine >2 mg/dL), and A—anemia (hemoglobin <10 g/dL.) [3]. The monoclonal proteins in blood and urine need to be investigated. If there is a presence of M protein in the blood, additional blood tests for calcium, uric acid, creatinine, and beta2 microglobulin are also recommended [3]. A bone marrow biopsy is needed. Investigations such as X-rays, computed tomography (CT) scans, and magnetic resonance imaging (MRI) are also recommended.

In 2014, the International Myeloma Working Group (IMWG) updated the diagnostic criteria for MM by adding three specific biomarkers that can be used for diagnosis in individuals without CRAB features, and these include “clonal bone marrow plasma cells ≥ 60%, serum free light chain (FLC) ratio ≥100 provided the involved FLC level is 100 mg/L or higher, or more than one focal lesion on magnetic resonance imaging (MRI)” [16]. In addition, they revised the definition to include computed tomography (CT) and positron emitted tomography—computed tomography (PET-CT)—to diagnose MM bone disease.

### 1.3. Clinical Features of Multiple Myeloma

According to the Mayo Clinic, the signs of MM include nausea, vomiting, constipation, mental confusion, loss of appetite, fatigue, feeling of tiredness, frequent infections, weakness in lower limbs, and feeling of excessive thirst [17]. Other signs and symptoms include infection, bleeding, renal insufficiency, lesions in the bone, fractures, fatigue due to anemia, and a decrease in the function of the immune system [3]. Bone pain may be felt in the lower part of the back, and often, pathological fractures are seen in the lower lumbar or pelvic region. Fractures due to focal lytic lesions, or generalized loss of bone, or elevated bone turnover due to excess cytokine production are frequently seen in MM patients [18].

### 1.4. Treatment of Multiple Myeloma with Various Drugs

For many years, melphalan, an alkalyting agent, prednisone, and other corticosteroids were used for the treatment of MM. In the last two decades, thalidomide, bortezomib, and lenalidomide were considered for the treatment of MM [19]. Proteasome inhibitor drugs such as Velcade (bortezomib) in combination with dexamethasone (the VD protocol) have been used for the treatment of MM. In the majority of cases, Velcade (bortezomib) and dexamethasone were combined with cyclophosphamide or adriamycin, or even with thalidomide (VTD), in order to improve the efficacy [20]. Bortezomib’s indication was expanded by the European Medicines Agency in August 2013 in order to include non-pretreated patients prior to planned high-dose treatment and stem cell transplantation [20]. In a few places, there is conventional treatment with conventional treatment of six cycles of melphalan/prednisone and lenalidomide [20]. In high-risk cases, combined autologous/allogeneic stem cell transplantation may prove to be beneficial. Table 1 summarizes all the different categories of drugs and their mechanisms of action in the treatment of MM.

## 2. Drug Resistance in Multiple Myeloma

It should not be forgotten that MM is still one of the most expensive cancers in terms of monetary costs [32]. Because multiple myeloma is so heterogeneous and complex, treating people with the same drug is often difficult. Hence, a proper understanding of the epigenetics and genetics of MM pathogenesis is important.

Initially, the patients become responsive to treatment, but at later stages they become resistant. This is the main challenge encountered with MM treatment. Many patients have relapsed, or, at times, treatment becomes refractory because of resistance. One of the processes underlying medication resistance is alteration in the myeloma cells’ adhesive abilities to the extracellular matrix or stromal cells in the bone marrow [33]. Researchers showed that genetic abnormalities and epigenetic aberrations that affect the patterns of DNA methylation and histone modifications of genes, mainly tumor suppressors, play a vital role in drug resistance in MM resistance [33].

The main causes for drug resistance are: (i) epigenetic alteration [34,35], (ii) genetic alteration [21], (iii) abnormal drug transport and metabolism, decreasing the intracellular drugs levels [36], (iv) apoptosis or other intracellular signaling pathways are dysregulated, and autophagy is activated [35], (v) cancer stem cells’ tenacity, which is resistant to many drugs and capable of self-initiating MM [35,37], (vi) the dependency of MM cells on stromal microenvironment components, which revealed a dysfunctional tumor microenvironment [35,38]. It has been reported that drug resistance in cancer can be acquired during therapy due to “selection pressure”, which is created by treatment, or it might even be caused by intrinsic processes in which malignant cells are resistant to drugs, even prior to treatment [39,40].

### Drug Resistance in Multiple Myeloma and Molecular Concept

The P-glycoprotein (P-gp), which is produced by the MDR/ATP-Binding Cassette (ABC) B1 gene, is one of the most critical players in drug resistance [21]. P-gp plays an important role in drug resistance when patients are treated with proteasome inhibitors, alkylating agents, and immunomodulatory drugs. The P-gp protein belongs to the pump transporter family, which mediates the cellular efflux of peptides and drugs. It was observed that 75% of patients treated with drugs such as doxorubicin, dexamethasone, and vincristine exhibited the expression of P-gp [41]. Interestingly, in untreated MM patients, P-gp expression was much less, giving evidence that the cumulative dose enhances P-gp expression in myeloma cells [21]. The myeloma cells express P-gp protein, which is directly related to the cumulative dose of vincristine and/or doxorubicin that the patients have received. However, patients treated with melphalan did not show an increase in P-gp expression, but interestingly, when the treatment included conventional drugs such as vincristine, doxorubicin, dexamethasone, bortezomib, and carfilzomib, there was an increase of P-gp expression [42,43]. Hence, P-gp may be considered as the protein candidate in all drug resistance cases. The inhibition of P-gp activity has received considerable attention in clinical studies. P-gp can also be a marker predicting the therapeutic response in MM.

The multidrug resistance associated protein (MRP) is also considered to be important for drug resistance. It is a member of the ATP-binding cassette superfamily of transport proteins. Other members of the ATP-binding cassette superfamily of transport proteins include BCRP (Breast Cancer Resistance Protein) and LRP (Lung Resistant Protein). Research studies have shown that the MRP gene can induce resistance to numerous drugs, such as doxorubicin, daunorubicin, and vincristine [44]. Multi-drug resistance refers to a state where the cancer cells become resistant to a wide variety of structurally and functionally unrelated drugs after exposure to a single chemotherapeutic agent [45].

## 3. Epigenetic Dysregulation in Multiple Myeloma and Future Planning

### Involvement of miRNAs and its Dysregulations and Outcome

According to research reports, miRNA can play an important role in the regulation of drug response in MM [35]. miRNAs can control various genes and regulate different cellular pathways. miRNAs can function as oncogenic or suppressors of tumors. It has been reported that miRNAs may control the drug response of MM cells through regulation of the apoptotic or proliferative pathways, which include p53 [46], which is a transcription factor that can stimulate tumor suppressor miRNAs or even inhibit some oncomiRNAs [47]. The miRNAs activated by p53 have been reported to affect the antiapoptotic genes, thereby enhancing the tumor suppressor activity of p53 [47].

Many miRNAs are observed to be dysregulated in the pathogenesis or drug resistant cases of MM. Upregulation of miR-21 has been reported to cause inhibition of apoptosis and increase drug resistance [48]. Upregulation of miR-221/222 leads to inhibition of apoptosis and modulation of drug influx–efflux and ABC transporters [49]. Upregulated miR106b-25 cluster, miR-181a/b, and miR-32 targeted genes were reported to control p53 activity in MM [50]. Downregulation of miR-631 involved the UbcH10/MDR1 pathway, which was associated with the development of BTZ resistance in myeloma cells [51]. Downregulation of miR-30c caused activation of the oncogenic Wnt/β-catenin/BCL9 pathway and promoted MM cell proliferation and drug resistance [52]. The allelic imbalances or heterozygosity loss were observed to be significantly associated with the altered expression of miRNAs, which were located in different regions, such as et-7b at 22q13.31 or miR-140-3p at 16q22. [53]. The use of antagomirs or miRNA mimics has been shown to possess anticancer activity and to have synergistic effects with antimyeloma drugs [54].

Epigenetic means changes in the expression of genes and chromatin structure without any change in the sequence of the DNA. Epigenetic dysregulation has been linked to many cancers. Hence, epigenetic modifications could be promising targets for future drug discovery.

DNA methylation, histone post-translational modifications, chromatin remodeling, and the silencing of gene expression have been implicated in the genesis and progression of MM. Dysregulation of DNA methylation has received considerable interest from different researchers. The expression of lineage-specific gene sets regulates the lymphopoiesis process. Progenitor cells have various ways of controlling gene expression, and there is a significant amount of control by epigenetic modifications. One of the most important epigenetic markers is DNA methylation, and this influences genetic architecture and gene expression [55]. The abnormal methylation of DNA is a typical feature of cancer cells. Hypomethylation is linked to genomic instability [56], whereas local hypermethylation is thought to play a role in tumor suppressor gene suppression [57]. Regarding the prognosis of MM, epigenetically-inhibited tumor suppressor genes could be considered very important. In most malignancies, genome-wide hypomethylation is accompanied by a global rise in entropy [58]. The MM epigenome is well characterized by substantial intra-tumor heterogeneity and methylation patterns that are primarily stochastic [59]. Research studies have shown that DNA methylation gains are associated with increased heterogeneity, and the same results were seen in acute myeloid leukemia [60]. GPX3, RSBP1, SPARC, and TGFBI have all been found to be epigenetically inactivated in MM samples, and their methylation status was linked to overall survival, giving evidence of their prognostic significance [55]. Currently, DNA methyltransferase inhibitors (DNMTis) are employed to correct abnormal DNA methylation patterns [55]. Interestingly, 5-azacytidine (azacytidine) and 5-aza-2′deoxycytidine (decitabine), two cytidine analogs, have been found to exhibit anti-myeloma activity, as evidenced by increased DNA damage, cell cycle arrest, and induction of MM cell death [61]. According to the latest research studies, enhanced stochastic methylation variation permits tumor cells to better adapt and discover novel paths in response to changes in the environment or treatment pressure [59].

Post-translational modifications (PTMs), such as acetylation, can occur on histones. Acetylation of histones is one of the most well-studied PTMs, making it a key participant in chromatin structural remodeling and gene transcription modulation [62]. Acetylated histones also serve as binding sites for proteins with bromodomains, which frequently control gene expression in a favorable manner [63]. Coding mutations involving histone acetyltransferases (HATs) are observed in MM. The main function of the HATs is to acetylate newly translated histones. There are two types of enzymes, type A HATs and type B HATs, which play a role in transferring acetyl groups to other proteins, such as oncogenes and tumor suppressors such as MYC, P53, and PTEN, changing their protein activities [64]. Histone deacetylase catalyzes the removal of acetyl groups, which results in transcriptional repression [65]. In the future, enzyme-binding acetylated histones may also prove to be beneficial in planning therapeutic options.

## 4. Role of Natural Products in Epigenetic Regulators

### 4.1. DNA Methyltransferase (DNMTs)

A few drugs target DNMTs, and they include well-known epigenetic drugs such as azacytidine and decitabine, which have been widely used as epigenetic modulators. The causes of concern are the adverse effects of azacytidine and decitabine, which include toxicity and poor chemical stability [66]. In this regard, the phytochemicals isolated from plants were found to be less toxic. There is a need to try phytocompounds present in natural products. A few natural products that act as DNMTis are: (i) Epigallocatechin-3-gallate (EGCG), (ii) curcumin, (iii) Kazinol Q, (iv) Nanaomycin, (v) Parthenolide, and (vi) Antroquinolol D [67].

### 4.2. Epigallocatechin-3-gallate (EGCG)

Epigallocatechin-3-gallate (EGCG) is found in green tea. Green tea, as a natural product, is also rich in polyphenols and may be beneficial in treating cancers. EGCG and polyphenols were reported to have chemoprotective effect against various cancers [68]. The first research on EGCG, published in 2003, showed that EGCG inhibited DNMT activity with an IC_50_ of 20 µM and reactivated methylation-silencing genes in cancer cells [69]. The same study also supported the view that inhibition of DNA methylation by any dietary constituent could be beneficial for the prevention of cancer. EGCG may also influence apoptosis, thereby proving its role in cancer.

### 4.3. Curcumin

The rhizome of *Curcuma longa* has several active compounds, and polyphenol non-toxic curcumin is one of them. The yellow pigment substance forms a major part of the spice (known as turmeric) used in India and Southeast Asia. Traditionally, curcumin has been used for the treatment of various diseases due to its anti-inflammatory, antibacterial, anti-allergic, and antioxidant properties. Curcumin can act as an epigenetic regulator. A recent study highlighted the anti-proliferation, anti-angiogenesis, and anti-mutation properties of curcumin and the apoptosis induced by it in cancer cells, thereby combating multidrug resistance [70]. The same published article reported the action of curcumin on various signaling pathways and cell cycle checkpoints. Interestingly, curcumin could boost myeloma cells’ chemosensitivity to Bortezomib, and this could be due to Notch1 suppression [71]. The polyphenol curcumin has been reported to inhibit the activity of histone acetyltransferase, which leads to histone acetylation inhibition [72]. Curcumin has been shown to induce methylation of the mTOR promoter through the upregulation of DNMT3A and DNMT3B [73]. Epigenetic regulation of the mammalian target of the rapamycin (mTOR) gene plays an important role in MM, and treatment with curcumin showed downregulation of mTOR to be associated with hypermethylation of its promoter [73]. Hence, curcumin could be used as an effective epigenetic modulator.

### 4.4. Kazinol Q

Kazinol Q, a natural chemical derived from the Formosan plant Broussonetia kazinoki, has been demonstrated to be effective as a DNMT1 inhibitor [74]. DNA methylation plays an important role as an epigenetic regulator of the transcription of various cancer genes. Kazinol Q inhibits DNMT activity. Kazinol Q may inhibit the proliferation of different cancer cells. Earlier reports showed that Kazinol Q inhibits the growth of MCF-7 breast and LNCaP prostate cancer cells, and such an action was performed through apoptosis induction [74]. Hence, Kazinol Q may be effective in the epigenetic regulation of MM because it has an effect on DNMT inhibition.

### 4.5. Nanaomycin A

Nanaomycin A is another quinone antibiotic that was discovered in a Streptomyces culture. The antibacterial activity of nanaomycin A is due to its ability to generate O_2_^−^ [75]. Nanaomycin causes the induction of apoptosis. The anti-proliferative action of Nanaomycin A was attributed to its selective inhibition of DNMT3B with an IC_50_ value of 500 nM [67]. Nanaomycin A showed selectivity for DNMT3B in biochemical assays and, according to researchers, this was the first identified DNMT3B-selective inhibitor that induced genomic demethylation [76].

### 4.6. Parthenolide

Parthenolide is an extract from the plant *Tanacetum parthenium*. In traditional medicine, the plant was used for the treatment of migraines and rheumatoid arthritis [77]. The plant was used for its anti-inflammatory and epigenetic cancer therapy properties [78]. The plant extract possesses anticancer properties. Parthenolide causes apoptosis and suppresses the growth of human 786-O kidney cancer cells [79]. The suppression of the transcription factor nuclear factor-kappa B is linked to apoptosis (NF-kB) [77]. Parthenolide epigenetically regulates the NF-kB target genes. Parthenolide, being an NF-kB inhibitor, can be effective as an anti-proliferative agent, which can induce apoptosis and check metastasis.

### 4.7. Antroquinolol D

Antroquinonol D, an ubiquinone derivative, is isolated from the mycelium of *Antrodia camphorate* [80]. According to research reports, Antroquinolol D was reported to inhibit the activity of DNMT1 in MDA-MB-231 breast cancer cells, having an IC_50_ value lower than 5 µM nM [67]. Hence, Antroquinolol D inhibits breast cancer growth and migratory potential, and such action is performed by inducing DNA demethylation and the recovery of several tumor suppressor genes. The plant product may also suppress NF-κB signaling.

### 4.8. Resveratrol

Resveratrol has been studied in multiple cancer cell lines, including MM, liver, skin, breast, prostate, lung, and colon. Resveratrol affects various signaling pathways that control cell division, growth, apoptosis, angiogenesis, and tumor metastasis [81,82]. Talib et al. described the tumor microenvironment being targeted by resveratrol that included reactive oxygen species (ROS), tumor associated macrophages, indoleamine 2,3-dioxygenase in dendritic cells, vascular endothelial growth factor (VEGF), tissue fibrosis, and interleukin 6 (IL-6) [83]. Resveratrol’s main chemotherapeutic action is apoptosis, which is associated with the activation of the tumor suppressor p53 as well as the induced activation of the Fas/CD85/APO-1receptor in various cancer cells [84,85]. Focusing on MM, resveratrol is found to be able to sensitize carfilzomib-induced apoptosis via promoting oxidative stress [86]. Resveratrol also induces apoptosis and obstructs the proliferation of MM cells by inhibiting the constitutive activation of NF-κB through abrogating the IκB-α kinase activation, subsequently downregulating survivin, cIAP-2, cyclin D1, XIAP, Bcl-xL, Bfl-1/A1, Bcl-2, and TNF-α receptor-associated factor 2 (TRAF2) [87].

Despite the fact that this compound attracts attention in cancer studies including MM, its efficacy in clinical studies is still limited. The high concentrations of resveratrol utilized in preclinical studies seem difficult to achieve in clinical settings [88]. However, its notable potential must not be taken lightly. Hence, more clinical trials are needed to prove its efficacy. Resveratrol acts on the enzymes that catalyze DNA methylation and histone modifications. The epigenetic modification is performed mainly through methylation and acetylation.

### 4.9. Quercetin

Quercetin is an important bioflavonoid that can be found in plants such as berries (blueberries and cranberries), apples, green leafy vegetables, onions, broccoli, cauliflower, cabbage, and nuts [89]. Interestingly, quercetin has the ability to neutralize free radicals and bind to metal ions. In MM, quercetin inhibits its cell proliferation by downregulating the expression of IQ motif-containing GTPase activating protein 1 (IQGAP1) and activating the extracellular signal-regulated kinase 1/2 (ERK1/2). Besides that, this compound is able to inhibit the activation of mitogen-activated protein kinase (MAPK) and the interaction between IQGAP1 and ERK1/2 in MM cells [90]. Modulation of DNA methylation and histone acetylation are the two important mechanisms by which quercetin acts as an epigenetic modifier.

### 4.10. Genistein

Genistein is predominantly found in Leguminosae, and it belongs to the aglycone subgroup of isoflavones. This compound’s anti-cancer and epigenetic regulation effects are centered on its ability to induce programmed cell death, increasing anti-cancer efficacy and inhibiting angiogenesis [91]. Treatment utilizing genistein is described as being able to inhibit human MM cell proliferation by upregulating miR-29b. Genistein is also able to inhibit NF-κB expressed by MM cells, Akt phosphorylation, and MM cell proliferation. Furthermore, it is also able to induce apoptosis in MM cells and decrease the expression of NF-κB-regulated genes [90].

## 5. Lifestyle Factors Concerned with Epigenetic Effects in Multiple Myeloma

Lifestyle is defined as the ‘distinguished way of life of an individual or group of people’. Lifestyle factors that generally influence health include diet, psychosocial and physical activities, working conditions, and smoking and addictions. Individual genetic profiles are intertwined with lifestyle and environmental factors, thus affecting their health status [92]. Epigenetic mechanisms are flexible genomic parameters that influence genome function under exogenous influence. Therefore, recent studies focus on changes in mechanisms such as DNA methylation, histone modifications, and microRNA expression that are influenced by lifestyle [93].

Amodio et al. stated that several DNMT, histone methyltransferase (HMT), and miRNA inhibitors are emerging as promising cancer management tools that are on the verge of clinical success [94]. One cannot forget the gene expression studies that are beneficial for molecular classification, for example, comparative genomic hybridization to map abnormalities for tumors [95]. The reversibility of epigenetic mechanisms has prompted current studies to understand their relationship with lifestyle in order to restore epigenetic equilibrium [94]. Figure 1 illustrates the targeted role of epigenetics in MM progression.

### 5.1. Diet

Dietary factors have an important role in the normal physiological function of the human body and are also involved in the process of pathological progression. The epigenetic variation of MM is dependent on dietary factors. As a result, developing strategies that use dietary compounds to target epigenetic modifications is critical in the prevention and treatment of MM [96].

Caloric restriction (CR) is the term describing a reduction in calorie intake by 10% to 40%, without causing any malnutrition. There is a scarcity of epigenetic data on the effects of pure CR in humans, as it is a difficult intervention to implement in the long run [97]. Two important trials, the DIOGENES and RESMENA studies, showed that CR significantly reduced inflammation and improved metabolic syndrome features [96,98,99]. To date, studies exploring epigenetically mediated changes in gene expression linked to CR have focused on aging processes. DNA methylation and histone modification are the main genetic codes that play significant roles in regulating the chromatin structure and gene expression in response to CR. However, studies utilizing CR intervention in describing the epigenetic changes among MM or general cancer populations are still scarce [100,101].

The consumption of fatty acids (FA) was found to have a significant impact on DNA methylation (either hyper or hypomethylation), acetylation or deacetylation of histones, and the miRNAs responsible for the repression or activation of genes. The desirable effects of disease prevention, including cancer progression, are associated with certain types of FA, such as n-3 PUFA (e.g., EPA-DHA) and MUFA (e.g., OA, palmitoleic). n-3 PUFA supplementation is associated with modification in DNA methylation profiles of blood leukocytes related to pathways involving inflammatory and immune responses. Besides that, in vitro studies revealed that the MUFA (oleic acid) anti-inflammatory effect was associated with DNA methylation signatures [96]. On the other hand, dietary FA types such as n-6 PUFA, saturated fatty acids (stearic and palmitic), and trans fatty acids (elaidic) are associated with DNA hypermethylation of the PPARγ1 gene promoter, a key factor in the activation of pro-inflammatory mediators and insulin resistance in macrophages [102,103]. These laboratory findings are in accordance with human studies that associate the intake of omega-3 essential fatty acid (n-3) with a reduced risk of MM, whilst a high intake of saturated fat is associated with poor MM progression [104,105].

Dietary polyphenols are supplied by the daily intake of fruits and vegetables. These plant polyphenols supply the source of various classes of polyphenols, including stilbenes, flavonoids, benzoquinones, phenolic acids, acetophenones, xanthones, and lignins. Preventive mechanisms that have been heavily studied are the capabilities of polyphenols to alter cancer cells’ epigenomes by remodeling their chromatin or reactivating silenced genes. Furthermore, previous studies have shown that polyphenols gained from daily diet intake are potential chemo-preventive agents due to their ability to modify histone and inhibit DNMTs. Common sources of food that contain polyphenols with protective roles against cancer are described in Table 2.

The ability of pharmacologically-dosed ascorbic acid (vitamin C) to either prevent or promote cancer progression may have contradictory evidence. However, increasing the intake of foods high in vitamin C may have a beneficial effect on chemo-prevention. Vitamin C has been reported to regulate the demethylation of DNA and histone, and it has a potential epigenetic impact on cancer prevention and treatment [112]. However, intake of certain natural products and antioxidants such as vitamin C may interact with the MM medication bortezomib, hence antagonizing its effect. Therefore, it is wise to highlight the potential negative interactions between patients’ anticancer therapies and complimentary therapies for MM patients in order to maximize the benefits of treatment [113].

### 5.2. Physical Activity

Multiple studies, including cohort studies, describe the association between high BMI in early adult life and the incidence of MM, suggesting that physical activity (PA) has either a direct or indirect effect on the progression of MM [114]. It has been found that exercise may reduce the risk of cancer through increasing global DNA methylation (DNMT1, DNMT2, DNMT3A, DNMT3B, DNMT3L) and influence the expression of genes related to muscle work, which includes PPARGC1A encoding peroxisome proliferator-activated receptor gamma coactivator 1-alpha (PGC-1α) [115].

The anti-inflammatory effect of PA through epigenetic regulation depends on the type of activity, endurance, body composition, gender, and age. Studies found that PA is associated with a desirable effect on the methylation of the ASC gene, plasma IL-1β levels, and NFKB2 gene promoter methylation. However, it is worth highlighting that overexertion of PA may also induce inflammation [96].

Elucidating the effect of PA on epigenetic regulation must include its interaction with diet, body composition, and age-related factors. These confounding factors must be discussed together in order to highlight the various outcomes of NFKB1 and NFKB2 gene methylation and inflammatory miRNAs [96,116]. Published data on MM and PA, which could be used to further evaluate the epigenetic regulation that is affected, is still scarce. Therefore, it is recommended to design a future study that focuses on the impact exerted by PA on the progression of the disease in order to understand its potential as part of the therapeutic regimen.

### 5.3. Addiction

Addiction to drugs and other addictive substances may lead to cellular and molecular changes, with many common neuronal changes in gene expression, as demonstrated by numerous human studies and animal models [117]. Some addictive drugs have been observed to increase the production of reactive oxygen species (ROS), and this results in oxidative stress conditions that eventually cause alteration in mitochondrial and nuclear gene expression [118].

Wong et al., early on, suggested that acute drug use produces enduring modifications in gene expression through epigenetic changes that affect susceptibility to addiction [119]. Increased vulnerability to substances of abuse will then feed back into an enhanced risk of subsequent drug use that brings about further alterations to the epigenome and gene expression.

Recent evidence with regard to opioid addiction points towards the substance promoting increased levels of permissive histone acetylation and decreased levels of repressive histone methylation, including alteration of the pattern of DNA methylation and noncoding RNA expression in the reward circuitry of the brain [120]. Nestler et al. reported that excessive dopamine signaling during drug use modulates gene expression and changes synaptic function and circuit activity [121]. A study in an experimental model of rats suggested that methamphetamine-induced alterations in global gene expression in the nucleus accumbens might be related, in part, to methamphetamine-induced changes in histone acetylation, and that may be secondary to changes in histone acetyltransferase and histone deacetylase expression [122].

A microarray analysis in a zebrafish model identified 1362 genes with a significant change in expression between control and those treated with nicotine or ethanol, with 153 genes shared by both ethanol- and nicotine-treated animals [123]. Ponomarev et al. reported that global changes in gene expression in the brains of alcoholics are primarily caused by chronic alcohol abuse, which eventually changes gene expression via alteration in chromatin states [124]. Renthal and Nestler have reported that chromatin structure regulation through post-translational modifications of histones (e.g., acetylation) is an important mechanism to translate environmental triggers, which include substances of abuse, into specific modifications of gene expression [125]. However, the association between substance addiction influencing epigenetic factors in MM is yet to be clearly elucidated.

### 5.4. Pollutants

Although the exact etiology of multiple myeloma still remains unclear, environmental factors have been suggested to increase the risk of the disease [126]. In particular, studies have shown that exposure to pesticides and Agent Orange, a defoliant chemical used in the Vietnam War, is associated with an elevated risk of MM, especially among workers in the agricultural industry [127]. Several other occupational and toxic agents have been implicated as potentially causative, yet the precise association of the disease through epigenetics is yet to be defined. Numerous research findings, however, show that exposure to heavy metals and environmental pollutants can interact with epigenetic factors such as genomic DNA methylation, specific gene promoter DNA methylation, DNA methyltransferase activity, global and gene-specific post-translational histone modifications, and alterations in microRNA expression levels [128].

Shukla et al. have demonstrated that particulate matter from air pollution disturbs the mitochondrial machinery and results in epigenome disruption. This epimutation is thought to involve mitochondrial DNA, allele imprinting, histone withholding, and non-coding RNAs, and the harmful changes may be passed down across generations, resulting in transgenerational epigenomic inheritance [129]. In addition, respiratory cell lines and animal models have also demonstrated distinct gene expression signatures in the transcriptome when exposed to particulate matter or ozone [130]. Particulate matter such as diesel exhaust particles and other environmental toxins also alter the expression of microRNA (short non-coding RNA that regulates gene expression), subsequently altering cellular processes leading to disease pathogenesis [130,131].

Industrially, occupational exposure to benzene gives rise to hematopoietic malignancy, possibly via its genotoxic action. The acquired epigenetic modifications may participate in benzene leukemogenesis, as benzene affects nuclear receptors, inciting post-translational alterations at the protein level that interfere with the function of regulatory proteins, which comprise oncoproteins and tumor suppressor proteins [132]. Other principal environmental and industrial pollutants, such as cadmium, chromium, lead, methyl mercury, arsenic, nickel, aluminum, benzene, polybrominated diphenyl ethers, polycyclic aromatic hydrocarbons, bisphenol A, and volatile organic compounds, have all been linked to epigenetic changes and are associated with a broad array of adverse health outcomes [133].

## 6. Epigenetic Treatment to Overcome Drug Resistance

Genetic mutations and epigenetic alterations were found to be responsible for the growth of the tumor and chemotherapy resistance [134]. It is always possible to reverse the changes with epigenetic modifications, and the malignant cells may be reverted to their normal state, thereby making them a suitable therapeutic target for appropriate drugs [135]. In the future, genomic technology could be used to better understand many somatic mutations. Many target genes of miRNAs encode for proteins that are involved in survival, proliferation, and drug resistance, and these areas could be studied in detail [136]. A few miRNAs are known to be direct transcriptional targets and tumor-suppressive effectors, and they could be potential areas of future drug targets.

A better understanding of histone modifications, such as deacetylation and methylation, could also be useful in combating drug resistance. Although many dysregulations of the epigenome in MM have been studied by researchers, only one epigenetic treatment for MM has been approved to date [137]. Pan-histone deacetylase (HDAC) inhibitor panobinostat is the only epigenetic drug that has been used to date [137]. This drug interferes with the chaperone function of HSP90 and leads to the suppression of tumor growth [138]. Panobinostat, when combined with a proteasome inhibitor and dexamethasone, showed better results with regard to survival in relapsing and refractory MM patients [138]. HDAC inhibitors have multiple modes of action, hence they could be useful in many drug resistance cases. HDACs alter histones and DNA, and also affect the post-transcriptional modifications of other proteins, which could perhaps act as an important epigenetic modifier [137]. Epigenetic inhibitors with a combination of miRNAs could be effectively used for drug resistant cases. Future clinical trials involving the epigenetic miRNA axis may also be beneficial.

## 7. Conclusions

There is a concern regarding the increase in the incidence of MM throughout the world. MM is a clonal plasma cell disorder. In this cancer, atypical plasma cells are found in the bone marrow. Proper diagnosis and early treatment of MM are necessary for better outcomes. The efficacy and tolerability of the drugs and the resistance they endanger will always remain the main challenges. Apart from the genetic abnormalities that are responsible for the etiology of MM, epigenetic aberrations are also responsible for the development of MM. Epigenetic drugs and modifications may help in combating the disease. In the present review, we discussed MM and modifiable epigenetics.

## Figures and Tables

**Figure 1 ijms-23-01649-f001:**
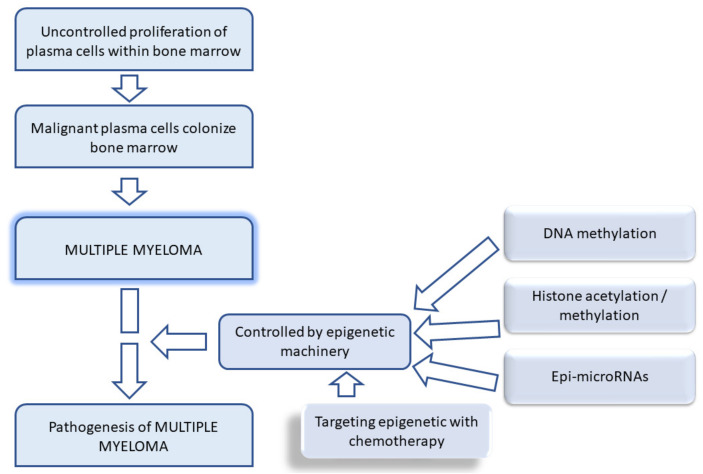
Schematic diagram showing etiology of multiple myeloma and the epigenetic machinery involved.

**Table 1 ijms-23-01649-t001:** Table showing different categories of drugs that are used to treat MM and their mechanisms of action.

Class of Drugs	Name of Drug	Mechanism of Action
Alkalyting agents	Melphalan; Cyclophosphamide	Formation of cross links between the two strands of DNA and impairment of DNA synthesis and cell replication [21]
Proteasome inhibitors	Bortezomib	Inhibits the ubiquitin mediated proteasome degradative pathway [21] Inhibitor of the NF-κB pathway Inhibits activations of β5 and β1 subunits of the 20S proteasome core in the 26S proteasome complex [22]
Immunomodulatory drugs (IMiDs)	Thalidomide; 2nd generation IMiDs such as lenalidomide and pomalidomide	Ubiquitin ligase modulations [21] Induce cell cycle arrest and apoptosis directly in MM cells [23]
Glucocorticoids	Prednisone; Dexamethasone	Induce apoptosis and alter the cellular cycle of malignant cells, interacting with the effects of IL-6 [24] Induce apoptosis through repression of transcription factor activity, and also inhibit the transcription of growth/survival genes [25]
Monoclonal Antibodies (mAbs)	Daratumumab	Targets the cell surface marker CD38, which is abundantly expressed on MM cells, and causes cellular death via a variety of immune-mediated pathways [26]
	Isatuximab	Binds selectively to CD38, thereby promoting MM cell death [27]
	Elotuzumab	Targets the CS1, which is a glycoprotein present on the surface of MM cells, also named signaling lymphocytic activation molecule family member 7 (SLAMF7) [28] Decreases MM cell adhesion to the bone marrow stroma and increases tumor cell death [29]
Histone Deacetylase Inhibitors (iHDACs)	Panobinostat	Increasing chromatin structure opening and, as a result, stimulating the expression of tumor suppressor genes [30]
Other drugs/karyopherin inhibitor	Selinexor	Tumor suppressor proteins are forced into the nucleus and activated, IκBα is trapped in the nucleus to suppress NF-κB activity, and oncoprotein mRNA translation is reduced [28] Induction of apoptosis in malignant cells, sparing normal cells [28]
Antibody drug conjugate	Belantamab mafodotin	Bind selectively to B-cell maturation antigen, together with monomethyl auristatin F (MMAF), a cytotoxic agent that mediates cell death [31]

**Table 2 ijms-23-01649-t002:** Table showing dietary sources of polyphenols and its epigenetic dietary compounds that affect cancer progression.

Dietary Sources	Epigenetic Dietary Compounds	Mechanism of Action
Garlic	Allyl mercaptan, Organosulfur	Histone deacetylases (HDAC) inhibitor [106]
Berries	Resveratrol	Dietary inhibitors of DNA methyltransferasses (DNMTs) HDAC inhibitor [107]
Nuts	Resveratrol Selenium	Dietary inhibitors of DNA methyltransferasses (DNMT) HDAC inhibitor [108]
Green tea	EC, ECG, EGC and EGCG	Histone acetyltransferase (HAT) inhibitor HDAC inhibitor miRNA modulator [109]
Turmeric	Curcumin	DNMT inhibitor miRNA modulator [70]
Beans	Genistein; Folate	DNMT inhibitor HDAC inhibitor miRNA modulator [110]
Green vegetables	Folate; Isothiocyanates; Sulforaphane	DNMT inhibitor HDAC inhibitor; miRNA modulator [111]

## Data Availability

Not applicable.

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
