# Peer review of "Multiple Myeloma: Challenges Encountered and Future Options for Better Treatment"

_ijms, 2022, doi:10.3390/ijms23031649_

Round 1

Reviewer 1 Report

Das et al summarizes in the manuscript entitled “Epigenetic regulation in Multiple Myeloma: an insight” some general aspects about multiple myeloma such as diagnostic criteria, clinical features, drugs used in the treatment of this disease and drug resistance, in addition to focus specifically on factors involved in epigenetic dysregulation in multiple myeloma, natural products with epigenetic effects and lifestyle factors related to epigenetic effects in multiple myeloma.

From my point of view, authors have to clarify and take into account several points:

-The title is very general and perhaps suggests that the review will mainly talk about molecular factors involved in epigenetic regulation in myeloma. However, epigenetic dysregulation in multiple myeloma is just a small section of the whole manuscript. I suggest thinking of a more specific title.

-English grammar should be generally revised in order to improve understanding of the text. Also, some scientific concepts need to be more precisely exposed. Some examples:

  • Line 14-15: “If detected early, the prognosis is good or else the survival rate falls, especially if distant metastasis has already occurred”. The meaning of this phrase is difficult to understand.
  • Line 43-44: “Researchers showed a positive association between body mass index (BMI) and the incidence of MM in whites, independent diabetes mellitus bring present”. Do the authors mean "being" where they write "bring"?
  • Lines 48-50: “A recent report published in 2020, researchers highlighted genetic and societal factors that contribute to racial and ethnic differences in the prevalence and prognosis of multiple myeloma and its precursor state”. Better: “In a recent report…”
  • Line 88-89: “For many years, Melphalan, an alkalyting agent, prednisone, and corticosteroids were used for the treatment of MM”. Better: “For many years, melphalan, an alkalyting agent, prednisone, and other corticosteroids were used for the treatment of MM”. Take into account that prednisone is also a corticosteroid.
  • Lines 90-91: “Proteasome inhibitor drugs such as Velcade (bortezomib) and dexamethasone (the VD protocol) have been used for the treatment of MM”. It is better to say: “Proteasome inhibitor drugs such as Velcade (bortezomib) in combination with dexamethasone (the VD protocol) have been used for the treatment of MM”
  • Lines 148-150: “Research studies have shown that DNA of the MRP gene can exhibit resistance to numerous drugs, such as doxorubicin, daunorubicin, and vincristine”. Maybe it is more precise to say: “Research studies have shown that MRP gene can induce resistance to numerous drugs, such as doxorubicin, daunorubicin, and vincristine”
  • Line 155: Do authors want to say “Involvement of miRNAs” instead of “Involvement on miRNAs”?
  • Lines 167-170: These two phrases are difficult to understand. Please, rephrase.
  • Lines 172-173: “Downregulation of miR-140-3p caused altered expression due to the occurrence of several allelic imbalances or loss of heterozygosity in 16q2 region”. Altered expression of what?
  • Lines 211-220: “Coding mutations involving histone acetyltransferases (HATs) are observed in MM. The main function of HATs is to acetylate newly translated histones. There are two types of enzymes, such as enzyme A and enzyme B, and they play a role in transferring acetyl groups to other proteins, such as oncogenes and tumor suppressors like MYC, P53, and PTEN, changing their protein activities [63]. Histone deacetylases catalyze the removal of acetyl groups, which results in transcriptional repression [64]. The promoter hypermethylation of many cancer-related genes, such as BNIP-3, p16, E-CAD, and DAPK-1, can be beneficial for knowing the prognosis of MM. In the future, enzyme binding acetylated histones may also prove to be beneficial in planning therapeutic options”. This whole paragraph is difficult to understand. Are “enzyme A” and “enzyme B” HATs? Do these enzymes acetylate oncogenes and tumor suppresor in addition to histones? Why do authors include in the middle of the paragraph a phrase regarding hypermethylation when they are talking about histone acetylation?
  • Line 234: “EGCG and polyphenols were reported to have chemoprotective against against various cancers”. It has to be said “EGCG and polyphenols were reported to have chemoprotective effects against various cancers”

-The two tables have to be better organized so that it is easier to distinguish what class each drug belongs to and what type of mechanism of action it has. The rows are intermingled with each other.

-In table 1, belantamab mafodotin, an antibody drug conjugate approved for the treatment of MM, should be included.

-In table 1, last row, I think that it should say "karyopherin inhibitors". Also, “Exportin 1 (XPO1)” does not have to be in the "name of drug" column since it is not a drug but a protein.

-In sections “4.6. Parthenolide”, “4.8. Resveratrol” and “4.9. Quercetin”, the epigenetic mechanism is not explained.

-In Figure 1, in the box where it says "histone acetylation" I think it would be better to write "histone acetylation / methylation". Also, in this Figure lncRNAs are mentioned, however nothing is said about lncRNAs in the manuscript.

-It is my personal feeling that sometimes the ideas in some paragraphs are not well structured and some sentences do not seem well linked.

-Minor comments:

  • Specify each acronym the first time it is mentioned and use it throughout the manuscript. For example, in line 55, MGUS is indicated. However this acronym has been already indicated in line 31.
  • In line 403, “PA” is mentioned. Authors have to specify what PA means.
  • Line 87: correct “mutliple”
  • Sometimes the authors used DNMTI and other DNMTi. Please, use always the same. The same for DNMTb and DNMT3B.

Reviewer 2 Report

The authors provide a great review of Epigenetic regulation in Multiple Myeloma. The coverage of topics is exhaustive. The subject matter covers the depth such that the readers will be greatly benefited. I recommend the publication of this manuscript in present form.

Reviewer 3 Report

The Authors described the epigenetic regulation in Multiple Myeloma. The review have a good impact in the field.

The review is very long to read, the descriptive part on the pathology could be reduced. The section of the mirna should perhaps be included in the section that should focus on the real purpose of the review which is epigenetics. The title does not reflect the review very much as the authors describe even more than a third of the impact of natural molecules on disease, diets etc, so it would be advisable to focus more on the text and the title. The drug resistance section should focus on the epigenetic factors that regulate drug resistance.

Round 2

Reviewer 1 Report

Authors have corrected the majority of the comments. Nevetheless, there are still some points to be addressed:

  • Line 179: In section “2.2. Involvement of miRNAs, its dysregulations and outcome” replace 2.2 with 3.1
  • Line 192: Replace miR106b~25 with miR106b-25
  • Table 1: the commentary “Targets the CS1, which is a glycoprotein present on the surface of MM cells, also named signaling lym-phocytic activation molecule family member 7 (SLAMF7) [28]” refers to elotuzumab, not to isatuximab as it is shown in the table. Isatuximab is an anti-CD38 mAb.
  • Also in Table 1, the reference regarding selinexor is the same as regarding panobinostat (reference 30). Please, correct it.
  • Line 302: maybe it is better to say “Parthenolide modulates the epigenetically regulated NF-kB target genes”
  • Line 244: In the phrase “A few drugs target DNMTIs, and they include well-known epigenetic drugs such as azacytidine and decitabine, which have been widely used as epigenetic modulators”. I think that DNMTIS should be substitued with DNMTs. Please, revise it.

Author Response

Line 179: In section “2.2. Involvement of miRNAs, its dysregulations and outcome” replace 2.2 with 3.1

Sorry for the error. We have rectified it.

Line 192: Replace miR106b~25 with miR106b-25

We have rectified it.

Table 1: the commentary “Targets the CS1, which is a glycoprotein present on the surface of MM cells, also named signaling lym-phocytic activation molecule family member 7 (SLAMF7) [28]” refers to elotuzumab, not to isatuximab as it is shown in the table. Isatuximab is an anti-CD38 mAb.

We have rectified it.

Also in Table 1, the reference regarding selinexor is the same as regarding panobinostat (reference 30). Please, correct it.

We have rectified the references.

Line 302: maybe it is better to say “Parthenolide modulates the epigenetically regulated NF-kB target genes”

We have rectified the sentence – “Parthenolide the epigenetically regulated NF-kB target genes.”

Line 244: In the phrase “A few drugs target DNMTIs, and they include well-known epigenetic drugs such as azacytidine and decitabine, which have been widely used as epigenetic modulators”. I think that DNMTIS should be substitued with DNMTs. Please, revise it.

We have revised it. It is mentioned as ‘DNMTs’

Reviewer 3 Report

The Authors have addressed all my comments.     

Author Response

No comments were given by reviewer 3. Hence, we assume that all corrections are acceptable.